# The Biological Behaviors of Neural Stem Cell Affected by Microenvironment from Host Organotypic Brain Slices under Different Conditions

**DOI:** 10.3390/ijms24044182

**Published:** 2023-02-20

**Authors:** Qian Jiao, Li Wang, Zhichao Zhang, Xinlin Chen, Haixia Lu, Yong Liu

**Affiliations:** 1Institute of Neurobiology, Environment and Genes Related to Diseases Key Laboratory of Education Ministry, Xi’an Jiaotong University Health Science Center, School of Basic Medical Sciences, Xi’an 710061, China; 2Department of Physiology, Shandong Key Laboratory of Pathogenesis and Prevention of Neurological Disorders and State Key Disciplines: Physiology, School of Basic Medicine, Qingdao University, Qingdao 266071, China

**Keywords:** neural stem cells, organotypic brain slice, microenvironment, biological behaviors, host

## Abstract

Therapeutic strategies based on neural stem cells (NSCs) transplantation bring new hope for neural degenerative disorders, while the biological behaviors of NSCs after being grafted that were affected by the host tissue are still largely unknown. In this study, we engrafted NSCs that were isolated from a rat embryonic cerebral cortex onto organotypic brain slices to examine the interaction between grafts and the host tissue both in normal and pathological conditions, including oxygen–glucose deprivation (OGD) and traumatic injury. Our data showed that the survival and differentiation of NSCs were strongly influenced by the microenvironment of the host tissue. Enhanced neuronal differentiation was observed in normal conditions, while significantly more glial differentiation was observed in injured brain slices. The process growth of grafted NSCs was guided by the cytoarchitecture of host brain slices and showed the distinct difference between the cerebral cortex, corpus callosum and striatum. These findings provided a powerful resource for unraveling how the host environment determines the fate of grafted NSCs, and raise the prospect of NSCs transplantation therapy for neurological diseases.

## 1. Introduction

Neural stem cells (NSCs) therapy has been considered a promising new strategy for stroke and other neurodegenerative diseases. NSCs could survive, proliferate and differentiate different types of neural cells after being transplanted, and consequently improve functional recovery from a stroke [1,2,3,4,5]. The amount and functional status of a desired type of cells in situ after injury became the key factors of neural functional reconstruction.

Increasing evidence has shown that NSCs’ development requires a specific niche to house NSCs and regulate their proliferation, differentiation and migration [6,7,8]. The niche regulatory signals can be cell–cell interactions, secreted factors and an extracellular matrix (ECM) [9,10,11]. From this perspective, cell cultures are limited by their defects to recreate the complex cell architecture and signaling properties of niche signals. Despite more evidence showing that NSCs transplantation therapy can repair damaged neuronal circuitry in the adult brain, the survival, differentiation, axon extension and synapse formation are dependent on the local condition. Recent research showed that the anatomical presynaptic inputs of human embryonic stem cell-derived midbrain dopamine or cortical glutamate neurons were largely dependent on the graft location [12]. Due to the various microenvironments after different brain injuries, such as stroke, hypoxia, trauma and neurodegenerative diseases, a detailed understanding of how the environmental signals that affect transplanted NSCs, such as inflammatory factors, cytoarchitecture and reactive gliosis, becomes even more important.

In this study, we engrafted primary cultured NSCs onto normal and damaged brain slices. In order to investigate the effects of the microenvironment on transplanted NSCs, we selected two types of brain injury: oxygen–glucose deprivation (OGD) and traumatic injury. We found that the survival and differentiation of transplanted NSCs were mainly influenced by host tissue-specific biochemical signals in situ, while the processes of extension and migration were mainly affected by the cytoarchitecture of the host tissue.

## 2. Results

### 2.1. The Biological Behaviors of NSCs-EGFP after Transplantation onto Brain Slices

Brain slices were cultured in DMEM/F12 (1:1) with 10% FBS for 14 days before use (Appendix A). The biological behaviors of NSCs-EGFP (Appendix A) were well-maintained after transfection, which has been detected before transplantation. Then, the survival and differentiation of NSCs in brain slices were further observed. NSCs survived well in brain slices for at least 30 days, as shown by the green fluorescence in Figure 1A. These cells differentiated into both astrocytes and different types of neurons after 14 days, as shown by EGFP^+^/glial fibrillary acidic protein (GFAP)^+^, EGFP^+^/ microtubule-associated protein 2 (MAP2)^+^, EGFP^+^/choline acetylransferase (ChAT)^+^, EGFP^+^/GAD67^+^ and EGFP^+^/tyrosine hydroxylase (TH)^+^ cells, respectively (Figure 1B).

To further analyze the effect of the specific niche signals from the brain slices on NSCs’ biological behaviors, the proliferation, differentiation and functional integration of NSCs were investigated and compared with that in cell culture systems and brain slices. The number of Ki-67^+^ cells increased in the transplantation system compared to the brain slices (Figure 1C), which meant the transplanted NSCs kept their proliferative ability. The expression of β-tubulin III significantly increased, whereas the expression of GFAP dramatically decreased 14 days after transplantation compared to the NSCs culture or brain slices (Figure 1D–G), which suggested that the microenvironment was more suitable for cell differentiation into neurons. In addition, synaptophysin protein expression was also detectable and distributed in (white arrows) or very close to (yellow arrows) EGFP-positive cells (Figure 1H), which indicated the potential synapse formation between the transplanted cells and hosts.

To further investigate the effect of niche signals from different regions of brain slices on NSCs biological behaviors, the survival, differentiation, migration and process extension of NSCs-EGFP in brain slices were investigated after they were transplanted onto the cerebral cortex or striatum, respectively (Figure 2A). More viable NSCs-EGFP were observed in the cortex than in the striatum, particularly one week after transplantation (Figure 2B). Those NSCs-EGFP developed into different morphologies, shown as typical uni- or bipolar, and the processes extended throughout the slice. Interestingly, the direction of NSCs-EGFP migration and process extension in the cerebral cortex, corpus callosum and striatum were distinctly different (Figure 2C–E). Correspondence with the original cytoarchitecture of host-tissue-transplanted cells presented radial processes in the cerebral cortex as shown by the NF200 and DCX staining, which were also consistent with astrocytes’ migrating direction. In the corpus callosum, transplanted cells’ processes were running parallel, whereas in the striatum, no consistent pattern was noticed. Taken together, these data suggested that transplanted NSCs responded differently to the different regional-specific guidance cues.

### 2.2. The Biological Behaviors of Transplanted NSCs on OGD-Damaged Brain Slices

An ischemic brain injury model was constructed by culturing the organotypic brain slices in OGD conditions (Appendix A). NSCs survived in brain slices for more than 30 days (Figure 3A,B). The number of NSCs shown by the fluorescence of EGFP dramatically decreased in the first several days and then maintained a low level of approximately 30% for the following days (Figure 3C). After 14 days of culture, the protein levels of nestin, β-tubulin Ⅲ and GFAP in different groups were compared and results showed that the expression of nestin was downregulated, whereas the expressions of β -tubulin and GFAP were upregulated in the group with NSCs transplantation (Figure 3D,E). Most of the NSCs differentiated into astrocytes, while some of them differentiated into different types of neurons, shown as EGFP^+^/ChAT^+^, EGFP^+^/GAD67^+^ and EGFP^+^/TH^+^ double-labeled cells (Figure 3E). Under the guide cues of brain slices, transplanted NSCs migrated differently in various brain regions (Figure 3G) and the potential synaptic connections were observed, shown by the expression of synaptophysin in the coculture system (Figure 3H).

### 2.3. The Biological Behaviors of Transplanted NSCs on Mechanically Injured Organotypic Brain Slices

Mechanically injured organotypic brain slices (Appendix A) were used to mimic traumatic brain injury. One day after the injury, NSCs-EGFP were transplanted (Figure 4A). NSCs survived on the host tissue for more than 30 days and the cells migrated to cover the damaged area (Figure 4B). After 14 days in culture, most of the transplanted NSCs differentiated into astrocytes (EGFP^+^/GFAP^+^). Very few of them were still nestin^+^. No obvious neuronal differentiation was observed. Interestingly, the EGFP^+^ cell’s process extension was obviously paralleling with the host cells, particularly GFAP^+^ astrocytes (Figure 4C). Along with the culture, transplanted NSCs intended to wire with the host cells and the axons continually extended (Figure 4D,E).

## 3. Discussion

The characteristics of NSCs, including self-renewal and multipotent differentiation, enhanced the superiority of NSC therapy in neurodegenerative diseases [13,14,15,16]. The cellular interactions between the host tissue and grafted NSCs are important prerequisites for the application of NSCs both in vitro and in vivo [17,18]. In the present study, we demonstrated that the biological behaviors of transplanted NSCs were affected by the host tissue environments both in normal culture conditions and after injury. Aside from the alteration of grafted NSCs’ survival and differentiation, the migration and process extension of NSCs’ daughter cells were guided by the regional-specific cytoarchitecture of host slices. These observations demonstrated the integration between the host environment and grafted NSCs that provided the potential guide cues for therapeutic benefits.

In the current study, exogenous NSCs were grafted to organotypic brain slices both in normal culture conditions and after injury. Compared with using animal models, which has some obstacles such as economy and ethics, the long-term cultured organotypic brain slices have many advantages, particularly the maintenance of regional-specific cytoarchitectural, neural circuits and neuronal activity, which make them a useful tool for the study of brain diseases [19,20,21,22,23,24,25]. The cocultured NSCs survived better and matured in a shorter time-frame compared with monocultures [23,26]. Furthermore, the platform was in favor of controlling and testing the changing environmental factors. Thus, the co-culture systems are a valuable tool for studying the influence of an environment on the development of NSCs.

Transplanted cells, with preferential regional adhesion related to the host architecture, were provided by mouse embryonic stem-cell-derived astrocytes and human neural progenitor cells seeded on hippocampal slices [23,27]. We found a distinct distribution of the transplanted NSCs in normal, OGD and traumatic brain slices. The transplanted NSCs survived in the location in which they stayed, then differentiated and migrated in the cortex and striatum with their style, which could be caused by the specific region’s host environment. Ischemia, hypoxia and traumatic injuries caused signal pathway activation, which could affect neural differentiation and maturation. Our study showed that the growth of NSCs decreased after the OGD was treated for 4 days. This evidence demonstrates that the injury environment is unfit for NSC survival and proliferation. Both in OGD and traumatic brain slices, more transplanted NSCs tended to be differentiated, especially astrocytes, which was different from that in normal conditions. Due to the rise in the subpopulation of reactive astrocytes that contribute to astrogliosis and scar formation, astrocytes play critical roles in functional recovery after a stroke and trauma [28]. Simultaneously, extracellular glutamine levels increased with astrocyte proliferation and activation, indicating the involvement of astrocytes in the conversion of glutamate to glutamine, which may be a benefit for functional recovery [29]. From these results, we supposed that more astrocyte differentiation would respond to these cues and contribute to impairment.

The migration and processes of transplanted cells were induced by the cues in the host special regions. Even though the brain slice was treated via OGD, the cues and architecture of the host tissue that affected the migration and process extension were still maintained. The tissue structure of traumatic regions changed and it was more attractive to transplanted cells for its radial process. The cerebral cortex is formed by the sequential radial migration of neurons generated in the subventricular zone. The extracellular matrix protein reelin, secreted by Cajal–Retzius cells in the marginal zone of the cortex, controls the radial migration of cortical neurons [30,31,32]. However, the interaction would be complex, with changes in cell morphology and extending processes throughout this region. Connexins, known for forming gap junctions, are required by the adhesion of migrating projection neurons to radial glia [33]. The process extension of NSCs in the radial pattern may also respond to these signals’ guiding. In the corpus callosum, graft-derived cells migrated from the grafting site along the corpus callosum past the midline, compared to the spinal cord white matter where graft-derived cells migrated from the injection site rostrally and caudally [34]. A study showed that callosal axonal segregation and pathfinding in the corpus callosum were mediated partly by the erythropoietin-producing hepatocyte receptor (Eph) A3 and Eph receptors [35,36]. The extension of processes through NSCs in parallel to the corpus callosum follows the fibers, and it may be that the NSCs responded to signals guiding this pathway. After the traumas were formed, the microenvironments were changed for pathological and metabolic events at cellular and molecular levels [37,38]. Meanwhile, the extracellular matrix of the injury region changed dramatically, which was important for cell survival, differentiation and migration, such as that of laminin, tenascins and fibronectin [39,40,41,42,43]. The extension of processes through transplanted NSCs was parallel to the radial processes of astrocytes in traumatic regions that might respond to the complex signal guidance [44,45,46]. It has been reported that astrocytes influenced NSCs niche not only by releasing and responding to diverse soluble factors but also expressed a wide range of extracellular matrix [47]. Additionally, NSCs transmigrated across the blood brain barrier model induced by glioma cells in vitro [48]. These evidences provided astrocytes as a guide to participate in the regulation of NSCs migration and process extension. In the present study, one striking alteration was astrocytes both in the OGD model and the traumatic model. The function changes of astrocytes might be induced by membrane receptors and ion channels, soluble signals and extracellular matrix, which are important factors affecting the behavior of NSCs. Thus, it is important to balance the migration and residence of NSCs in the target region, because overly extensive migration decreases the number of functional cells and would enhance the risk of disadvantageous effects in therapy application.

Early synaptic integration of transplanted NSCs is important for the survival and functional formation of new neurons. To initiate the formation of synapses, the terminal of the axon starts to branch into multiple tiny arbors when they reach the target area [49]. We found the extending processes of neurons had a bold terminal end that may be concerned with synapse formation. Because axon guidance molecules, such as ephrin, bone morphogenetic proteins (BMPs) and Wnts, can also regulate the localization and formation of synapses [49], the growing direction of processes moved toward the injury region as observed in our traumatic slice models. These data suggested that the post-injury environment is constantly evolving and influencing the maturation of transplanted NSCs. Moreover, the present study strongly proved that the organotypic slice culture system is a useful platform for evaluating the biological behaviors of transplanted NSCs affected by the host microenvironment.

## 4. Material and Methods

### 4.1. Prepare of NSCs

Pregnant Sprague Dawley rats were provided by the Experimental Animal Center, Xi’an Jiaotong University Health Science Center. One pregnant rat was housed per cage. All procedures involving animal work conformed to the ethical guidelines of the NIH Regulations for Experimentation on Laboratory Animals set out by Xi’an Jiaotong University (NO: 2018-280).

NSCs were isolated from the cerebral cortex of rat embryos on embryonic day 14 (E14) and cultured in a serum-free growth medium following the protocol of Fred Gage [50] and optimized in our lab [51]. The meninges of the brains from embryos were carefully removed. The cerebral cortex tissue was separated from the brain. Then, the cerebral cortex tissue was trypsinized, mechanically triturated into single cells and cultured in a serum-free growth medium. The NSCs growth medium contained DMEM/F12 (Dulbecco’s modified Eagle medium and Ham’s F12, 1:1, Cat#12400024), 10 ng/mL bFGF (Cat#13256-029), 20 ng/ mL EGF (Cat#PHG0311L), 100 UI penicillin (Cat#15140122), 100 μg/mL streptomycin (Cat#15140122), 1% N2(Cat#A1370701), and 2% B27 supplement (Cat#17504044) (all above regents were from Invitrogen, Grand Island, NY, USA) and 2.5 μg/mL heparin (Sigma, St. Louis, MO, USA). Cells were sub-cultured in suspension every 7 days and they formed neurospheres. Upon the passage, neurospheres were trypsinized and mechanically triturated into single cells and reseeded at 1 × 10^5^ cells /mL and cultured either in a growth medium or differentiation medium that was DMEM/F12 (1:1) with 1% fetal bovine serum (FBS, Invitrogen, Cat#10091148).

NSCs were labeled via recombinant lentivirus that carried *Egfp* before transplantation. The pLOV lentiviral vector prepared by Neuronbiotech (Shanghai, China) carries a CMV promoter that drives EGFP expression. The viral solutions (8 × 10^8^ TU/mL) were stored in aliquots at −80 °C. NSCs in passage 2 were processed for lentivirus infection. NSCs were seeded at a density of 1 × 10^5^ cells/mL 12 h before infection. The green fluorescence expressed by NSCs-EGFP was detected under an inverted fluorescence microscope (Leica, Wetzlar, Germany) 48 h after infection. At this point, parallel neurospheres were trypsinized into single cells and analyzed using a fluorescence-activated cell sorter (FACS BD Biosciences, Franklin Lakes, NJ, USA).

### 4.2. Culture of Organotypic Brain Slice

Organotypic brain slices were prepared following the previous protocol and optimized in our lab [52,53]. Newborn Sprague Dawley rats (provided by the Experimental Animal Center, Xi’an Jiaotong University Health Science Center) were anesthetized with sodium pentobarbitone (45 mg/kg) and decapitated on postnatal day (P) 3. The whole forebrain was cut into 200 μm thick coronal slices using a vibrating microtome (World Precision Instruments, Sarasota, FL, USA) and immersed in cold artificial cerebrospinal fluid (4 °C). Then, the slices were placed on the semi-permeable membrane of the Transwell (Millipore, Cat#PIHP03050, Darmstadt, Germany). The culture medium was composed of 10% FBS (Invitrogen, Cat#10100147, Grand Island, NY, USA) and 90% DMEM/F12 (1:1). Slices were routinely observed under a phase-contrast microscope at two-day intervals and half the medium was replaced every three days. After 14 days of being cultured in vitro, the brain slices were used for the following experiments.

### 4.3. Brain Injury Models

The OGD model was used to imitate cerebral ischemia. The inserts with brain slices were placed into different wells of six-well plates with DMEM (1 mL in each well, no glucose, Invitrogen, Cat#11966025, Grand Island, NY, USA) after three washes. The cultures were then placed into a chamber in an anaerobic workstation (Bugbox; Ruskinn Technology, Bridgend, UK) that was pre-filled with a gas mixture of 5% CO_2_, 0.3% O_2_ and 94.7% N_2_. The chamber was sealed for 0.5, 1 and 2 h at 37 °C. After OGD, slices were maintained in normal culture conditions for another 24 h before the further experiment. The cultures in the control group were maintained under a normoxic atmosphere in DMEM.

A mechanical trauma model was used to imitate traumatic brain injury. Trauma was constructed in organotypic brain slices after 14 days in culture. The fine end of the yellow tip (10 μL) was used to make minor scuffing randomly by straightly piercing the tissue to imitate mechanical trauma in vivo.

### 4.4. NSCs Transplantation

A volume of 10 μL cell suspension that contained 1 × 10^5^ labeled NSCs (NSC-EGFP) was added to the surface of the brain slices that had either been cultured in the OGD condition for 24 h or traumatically injured 1 day in advance. Then, the slices were maintained in a normal condition for another 14 to 30 days. The green fluorescence was detected 24 h after transplantation and then observed every 2 days for more than 30 days. Brain slices in different groups were fixed with 4% PFA at 4 °C for further experiments.

### 4.5. Immunostaining

Immunostaining was employed to characterize NSCs, neurons and astrocytes, utilizing mouse anti-nestin (monoclonal antibody, 1:200, Millipore, Cat#MAB353, Darmstadt, Germany), mouse anti-β tubulin III (monoclonal antibody, 1:200, Millipore, Cat#MAB5564, Darmstadt, Germany) and mouse anti- GFAP (monoclonal antibody, 1:1000, Millipore, Cat#MAB360, Darmstadt, Germany) following a standard protocol and optimized in our lab [51,53]. Rabbit anti-MAP2 (polyclonal antibody, 1:200, Sigma, Cat#M3696, St. Louis, MO, USA), rabbit monoclonal anti-DCX (monoclonal antibody, Abcam, 1:200, Cat#ab207175, Cambridge, UK), rabbit anti-TH (polyclonal antibody, 1:50, Abcam, Cat#ab6211), rabbit anti-Glutamic acid decarboxylase 1 (polyclonal antibody, GAD67, 1:1000, Abcam, Cat#ab97739, Cambridge, UK) and rabbit anti-ChAT (polyclonal antibody, 1:400, Abcam, Cat#ab223346, Cambridge, UK) were used to characterize dopaminergic, GABAergic and cholinergic neurons [53]. Polyclonal rabbit anti-Ki-67 (1:300, Millipore Cat#AB9260, Darmstadt, Germany) was applied to analyze NSCs’ proliferation [51].

After three washes with 0.01M PBS containing NaCl 9 g, Na_2_HPO_4_•12H_2_O 2.8998 g and NaH_2_PO_4_ •2H_2_O 0.2964 g (all from Hushi, China) that dissolved in 1000 mL of deionized water, the fixed cells and slices were incubated with a blocking solution containing 5% normal goat serum and 0.25% Triton X-100 in PBS for 2 h followed by the incubation with primary antibodies at 4 °C overnight. CY3-conjugated goat anti-mouse and goat anti-rabbit (Invitrogen, 1:500, Cat#A10522, Grand Island, NY, USA) secondary antibodies were applied on the following day. Cell nuclei were counter-stained with DAPI-containing mounting media (Beyotime, Cat#C1006, Nantong, China) and visualized under a fluorescent microscope (Olympus BX51, JPN, objective lens contained 10×, 20× and 40×) equipped with a DP70 digital camera and the DPManager (DPController, Olympus, Tokyo, Japan) software at 20 °C–26 °C. For the negative control, the primary antibody was replaced with 0.01M PBS.

### 4.6. Western Blot Analysis

Slices were lysed in a RIPA lysis buffer. The insoluble material was removed through centrifugation at 12,000 rpm for 15 min at 4 °C. Cell lysates were subjected to electrophoresis using 10% SDS polyacrylamide gels and transferred to nitrocellulose membranes. After being blocked in 5% non-fat dry milk in TBST for 2 h, the membranes were incubated with the primary antibodies at 4 °C overnight and the secondary antibody at room temperature for 2 h. Monoclonal mouse anti-nestin (Millipore, 1:1000, Cat#MAB353, Darmstadt, Germany), anti-β tubulin III (Millipore, 1:1000, Cat#MAB5564, GER), anti-GFAP (Millipore, 1:1000, Cat#MAB360, Darmstadt, Germany) and anti-β-actin (1:5000, Santa Cruz Bio, Cat#sc-8432, Dallas, TN, USA) were used. Blots were detected through chemiluminescence with the ECL method (Pierce, Cat#32106, Dallas, TX, USA) and the data were analyzed with Image J (version 1.61).

### 4.7. Quantification and Statistic Analysis

The fluorescence obtained from nine areas in three individual NSCs treated slices (three in each) was measured with Image-Pro Plus (IPP, Media Cybernetics, Rockville, MD, USA) software to reveal the biological behavior of transplanted NSCs. The length of the processes obtained from three individual NSCs treated slices were measured by IPP to reveal the process extension of transplanted NSCs. All processes were measured and the top 30 longest were selected and averaged. All data were analyzed with the SPSS 17.0 software and GraphPad Prism 8. The results from the value of fluorescence intensity in the cortex and striatum at different times were analyzed using two-way ANOVA, followed by Dunnett’s multiple comparisons test and *t*-tests. One-way ANOVA and Student’s *t*-test were used. A *p* value < 0.05 was considered significant.

## 5. Conclusions

The present study suggests diverse influences of the tissue environment on transplanted NSCs. The highlighted differences in the biological behavior of transplanted cells in injury regions emphasized the need for finding combined treatment with NSCs therapy. Furthermore, we identified diverse influences of the host environment on NSCs, which will affect the potential of transplanted cells to provide therapeutic benefits.

## Figures and Tables

**Figure 1 ijms-24-04182-f001:**
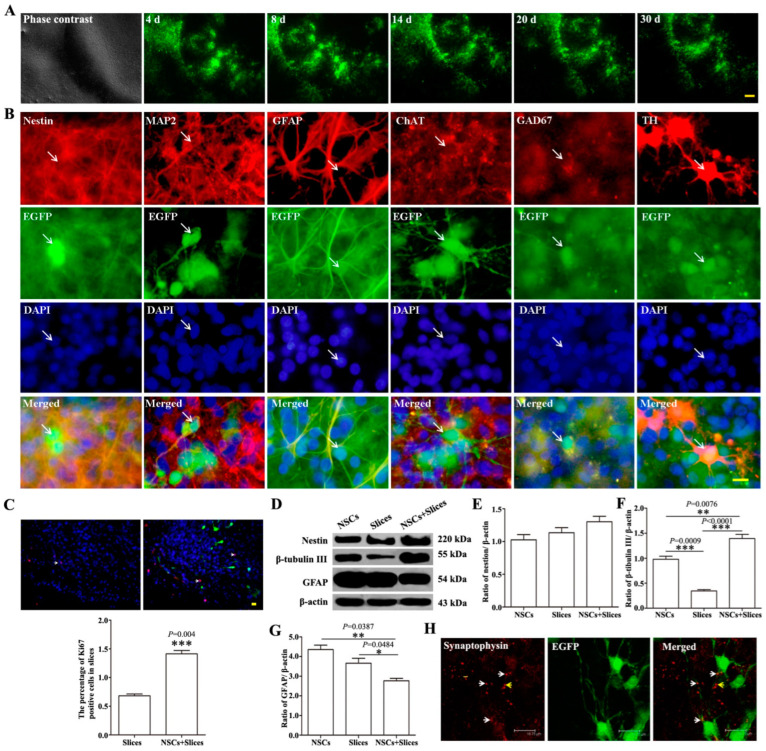
The survival and differentiation of transplanted NSCs on organotypic brain slices under normal conditions. (**A**): Transplanted NSCs survived on host brain slices for more than 30 days. Scalar bar = 500 μm. (**B**): Transplanted NSCs on host brain slices differentiated to astrocytes and different types of neurons. White arrows: the double stained cells. Scalar bar= 20 μm. (**C**): The number of Ki-67-positive cells in brain slices with NSCs transplantation was significantly higher than in the slices without transplantation. White arrows: the double stained cells. Scalar bar = 20 μm. Ki-67 was shown in red. The values are mean ± SE. *** *p* < 0.001, *n* = 5. (**D**–**G**): The protein levels of nestin, β-tubulin III and GFAP detected through Western blotting showed a dramatic increase in β-tubulin III and a decrease in GFAP in brain slices after NSCs transplantation. The values are mean ± SE. * *p* < 0.05, ** *p* < 0.01, *** *p* < 0.001, *n* = 5. (**H**): Potential synaptic connections between transplanted NSCs and host cells. White arrows: synaptophysin protein expressed in EGFP-positive cells; yellow arrows: synaptophysin protein expressed in host cells which very closed to EGFP-positive cells. Scalar bar = 18.75 μm.

**Figure 2 ijms-24-04182-f002:**
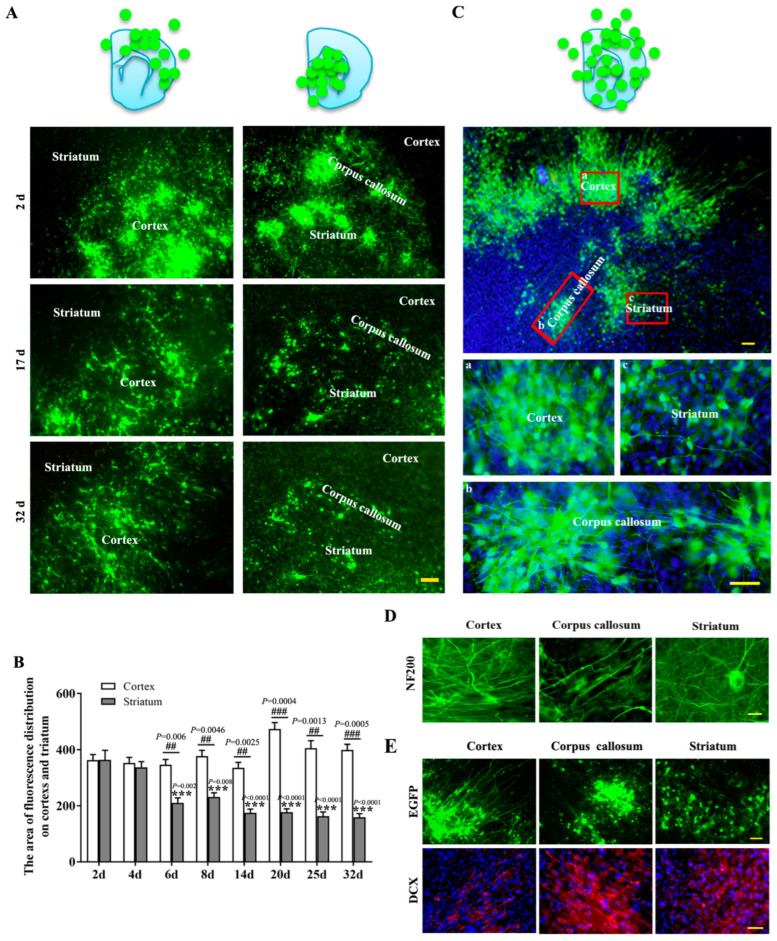
The survival and migration of transplanted NSCs on different part of host tissue. (**A**): The transplanted NSCs survived differently on cortex and striatum. Scalar bar = 50 μm. (**B**): Statistical analysis was a two-way ANOVA (Df = 7, F = 6.914, *p* < 0.0001, for time; Df = 1, F = 238.2, *p* < 0.0001, for age. Df = 1, F = 6.864, *p* = 0.0108, for genotype; Df = 4, F = 3.384, *p* = 0.0228, for age), followed by Dunnett’s multiple comparisons test among different time points, and *t*-tests performed between cortex and striatum at the same time points. The values are mean ± SE, *n* = 5. ^##^
*p* < 0.01, ^###^
*p* < 0.001; *** *p* < 0.001 in comparison with striatum 2 d. (**C**): The migration of transplanted NSCs on cerebral cortex, striatum and corpus callosum. Red box: the zoomed-in region. (a) Cortex; (b) Corpus callosum; (c) Striatum. Scalar bar = 50 μm. (**D**): The growth of neurites in cerebral cortex, striatum and corpus callosum. Scalar bar = 20 μm. (**E**): The distribution and migration of transplanted NSCs in different brain regions. DCX2^+^ -nerve fibers of host neuron showed in red. Scalar bar = 50 μm.

**Figure 3 ijms-24-04182-f003:**
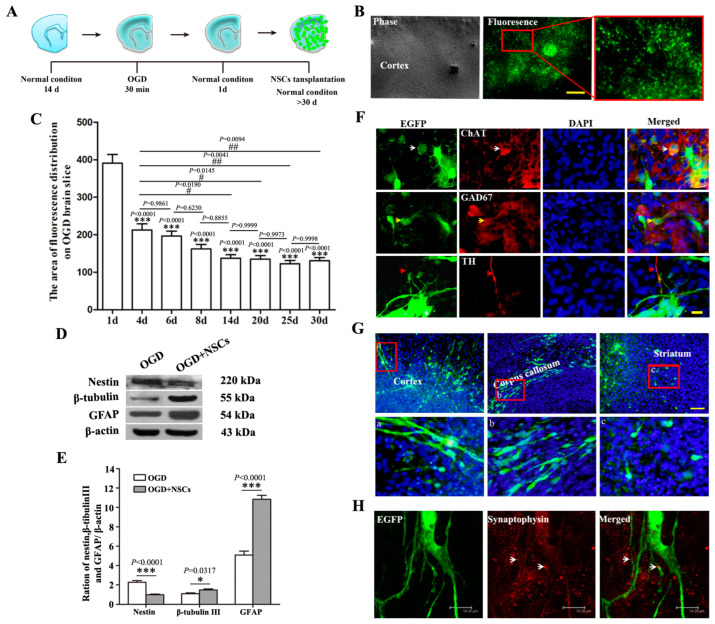
The biological behaviors of transplanted NSCs on host organotypic brain slices under OGD conditions. (**A**): The schematic diagram of transplantation of NSCs on OGD brain slices. (**B**,**C**): The survival of transplanted NSCs on cerebral cortex of brain slice after OGD treatment. Red box: the zoomed-in region. Scalar bar = 500 μm. The values are mean ± SE. *** *p* < 0.001 in comparison with 1d; ^#^
*p* < 0.05, ^##^
*p* < 0.01 in comparison with 4d, *n* = 5. (**D**,**E**): The expression of nestin, β-tubulin III and GFAP. The values are mean ± SE. * *p* < 0.05, *** *p* < 0.001, *n* = 5. (**F**): Transplanted NSCs differentiated to ChAT (white arrow)-, GAD67 (yellow arrow)- and TH (red arrow)-positive neurons. Scalar bar = 20 μm. (**G**): Different migration patterns of transplanted NSCs on cerebral cortex, striatum and corpus callosum. Red boxes: the zoomed-in regions. (a) Cortex; (b) Corpus callosum; (c) Striatum. Scalar bar = 100 μm. (**H**): The potential synaptic connections between transplanted NSCs and host cells. White arrows: synaptophysin protein expressed in EGFP-positive cells. Scalar bar = 18.75 μm.

**Figure 4 ijms-24-04182-f004:**
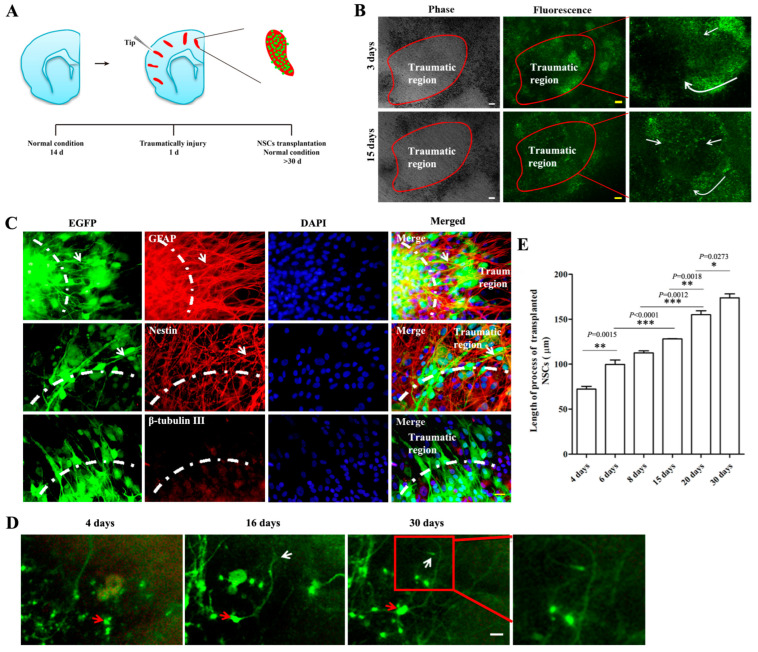
The biological behaviors of transplanted NSCs on traumatic injured brain slices. (**A**): The schematic diagram of NSCs transplantation to traumatic injured brain slices. Red regions represented the mechanical traumas and the green dots represented NSCs-EGFP. (**B**): Transplanted NSCs survived well and spread out (white arrows) to cover the damaged area (circled by red line). (**C**): Transplanted NSCs differentiation in injured host tissue. White arrow: the double stained cells. The boundary between the damaged area and the normal area showed by white dash line. (**D**): Process extension of transplanted NSCs. Red arrow: cell body; white arrow: growth cone. Red box: the zoomed-in region. (**E**): Elongation of cell process along with culture. Scalar bar = 100 μm. The values are mean ± SE. * *p* < 0.05, ** *p* < 0.01, *** *p* < 0.001, *n* = 5.

## Data Availability

Not applicable.

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
