# Peer review of "The Biological Behaviors of Neural Stem Cell Affected by Microenvironment from Host Organotypic Brain Slices under Different Conditions"

_ijms, 2023, doi:10.3390/ijms24044182_

Round 1

Reviewer 1 Report

The study entitled “The biological behaviors of neural stem cell affected by microenvironment from host organotypic brain slices under different condition” aimed to understand the biological behavior of transplanted neural stem cells. The authors performed engraftment of rat NSCs onto normal and damaged rat organotypic brain slices. They observed the impact of the host tissue-specific signals on the survival and differentiation of the transplanted cells throughout long-term organotypic slices culture. The migration of cells was pointed out to be affected by the host tissue cytoarchitecture. The big advantage of this study is the use of long-term organotypic brain slices culture, providing more useful data to study NSC behavior.

This paper seems to be interesting representing one of the hot research topics regarding the therapeutic potential use of NSCs for cell-based therapy in a plethora of neurodegenerative and neurotraumatic diseases.

We recommend the publication of this paper after doing the following changes:

1. There are several typos found throughout the article. Here are some examples:  

Line: 67-79: many spaces lacks

Line 126: lack of space after ‘cholinergic neurons’

Line 172: too big space

Line 242: Correct the Figure number

Line 331: lack of space after ‘matrix’

Line 339: lack of space after ’area’

...Please, check the manuscript again.

2. Please, provide the vendor’s name of the PBS. What kind of PBS (with or without ions) did you use?

3. Also, the figure quality is unfortunately low. For example, it is not possible to read the numbers and asterisks in Figure 1. The same for Figure 2B, Figure 4E....

4. In the discussion, we would like to ask the authors for providing more references, especially at the beginning of this section.

Author Response

The study entitled “The biological behaviors of neural stem cell affected by microenvironment from host organotypic brain slices under different condition” aimed to understand the biological behavior of transplanted neural stem cells. The authors performed engraftment of rat NSCs onto normal and damaged rat organotypic brain slices. They observed the impact of the host tissue-specific signals on the survival and differentiation of the transplanted cells throughout long-term organotypic slices culture. The migration of cells was pointed out to be affected by the host tissue cytoarchitecture. The big advantage of this study is the use of long-term organotypic brain slices culture, providing more useful data to study NSC behavior.

This paper seems to be interesting representing one of the hot research topics regarding the therapeutic potential use of NSCs for cell-based therapy in a plethora of neurodegenerative and neurotraumatic diseases.

Response: We appreciated very much for reviewer’s comments.

We recommend the publication of this paper after doing the following changes:

  1. There are several typos found throughout the article. Here are some examples:  

Line: 67-79: many spaces lacks

Line 126: lack of space after ‘cholinergic neurons’

Line 172: too big space

Line 242: Correct the Figure number

Line 331: lack of space after ‘matrix’

Line 339: lack of space after ’area’

...Please, check the manuscript again.

Response: We appreciated very much for reviewer’s comments. Following the kindest suggestion, we have gone through the whole manuscript and checked the written specifications  (English editing by MDPI, Certificate-60654). The typos including that mentioned above have been revised in this revision.

  1. Please, provide the vendor’s name of the PBS. What kind of PBS (with or without ions) did you use?

Response: We are sorry for the missing information. PBS (0.01M) used in the current study contained NaCl 9 g, Na2HPO4•12H2O 2.8998 g, NaH2PO4 •2H2O 0.2964 g (all from Hushi, CN) that dissolved in 1,000 mL deionized water. Above information has been added into the revised manuscript “5. Material and methods” and marked with blue color.

  1. Also, the figure quality is unfortunately low. For example, it is not possible to read the numbers and asterisks in Figure 1. The same for Figure 2B, Figure 4E....

Response: Many thanks for the suggestion. The figure quality has been improved and the size of numbers, asterisks, pound signs and P value characters in Figure 1, Figure 2B, Figure 3C, Figure 3E and Figure 4E have been enlarged for easier reading.

  1. In the discussion, we would like to ask the authors for providing more references, especially at the beginning of this section.

Response: Many thanks for your suggestion. Following relevant references have been provided in the discussion of revised manuscript.

New references in revised manuscript:

  1. Kahroba, H., et al., The role of Nrf2 in neural stem/progenitors cells: From maintaining stemness and self-renewal to promoting differentiation capability and facilitating therapeutic application in neurodegenerative disease. Ageing Res Rev, 2021. 65: p. 101211.
  2. Gordon, A., et al., Long-term maturation of human cortical organoids matches key early postnatal transitions. Nat Neurosci, 2021. 24(3): p. 331-342.
  3. Choi, S.H., et al., A three-dimensional human neural cell culture model of Alzheimer's disease. Nature, 2014. 515(7526): p. 274-8.
  4. Zhang, Y., et al., Three-dimensional-engineered bioprinted in vitro human neural stem cell self-assembling culture model constructs of Alzheimer's disease. Bioact Mater, 2022. 11: p. 192-205.
  5. Park, K.I., Y.D. Teng, and E.Y. Snyder, The injured brain interacts reciprocally with neural stem cells supported by scaffolds to reconstitute lost tissue. Nat Biotechnol, 2002. 20(11): p. 1111-7.
  6. Peruzzotti-Jametti, L., et al., Neural stem cells traffic functional mitochondria via extracellular vesicles. PLoS Biol, 2021. 19(4): p. e3001166.

Reviewer 2 Report

Present paper written by Jiao et al., with the title “The biological behaviors of neural stem cell affected by microenvironment from host organotypic brain slices under different condition” demonstrates the differential effect of tissue architecture of different brain regions on transplanted NPCs. The study is interesting, however the manuscript should be reviewed and clear up some confusing points before it can be accepted for publication.

·         The image quality overall is quite bad in the manuscript, although it is ok in the supplementary material! The graphs are not readable in the draft version. Please use higher quality versions.

·         Please review for the language, there are many typos and sentences with loss of meaning due to grammatical errors. In many cases the plural suffix is missing (for example in the title: …different conditions)

·         Line 30. Use neurodegenerative diseases, not „neural degenerative diseases”

·         Line 71. Are NPCs cultured in the form of neurospheres? Explain the method more clearly.

·         Line 104. Does the control group have medium with glucose?

·         Line 106. How did you make sure that all slices had similar mechanical damage? Did you make specific number of holes with the pipette tip?

·         Line 152. Process extension sounds strange, consider using another term such as neurite outgrowth

·         Co-localization of EGFP with some stainings in Figure 1.B is not clear. I do not know if it is from the low image quality in the draft, but if not would be good to put green and red channel also separately if not (such as MAP2, not clear if the transplanted cell is positive or is it from the slice)

·         Fig.1C – please indicate which brain region this is. Is the statement with Ki-67+ cells correct for all the brain regions?

·         When talking about NSC migration, (such as figure 3G) do the authors indicate the cells are migrating due to their shape, or is there an original location where the cells were transplanted and they were observed later outside of this location? When the cells are transplanted, they can also move on the surface of the slice due to the shape of the brain slice as the edges typically get thinner. Or the parts with mechanical damage can differ in topology, causing not equal distribution of the NPCs. Do the authors account for these? Zoomed out images such as Morgan et al., 2012; Fig3.A can be good to show there is no difference.

·         Fig.4C what does the dashed white line mean? Is it the area of injury? Please specify.

·         Line 277. Co-cultured NSCs survive better etc… Would be good to specify NSCs co-cultured with organotypic slices, as they can also be co-cultured with other type of cells.

·         Authors mention guidance cues many times, could be good to discuss in one/a few sentences what can be these, in which way they can different in the brain regions evaluated in this study.

Author Response

Present paper written by Jiao et al., with the title “The biological behaviors of neural stem cell affected by microenvironment from host organotypic brain slices under different condition” demonstrates the differential effect of tissue architecture of different brain regions on transplanted NPCs. The study is interesting, however the manuscript should be reviewed and clear up some confusing points before it can be accepted for publication.

  • The image quality overall is quite bad in the manuscript, although it is ok in the supplementary material! The graphs are not readable in the draft version. Please use higher quality versions.

Response: We are sorry for the fuzzy pictures. The difficulties for reading the images that presented in this manuscript may be due to the document type. All graphs have been double checked to make sure that each figure meets the requirements. The original figures have also been uploaded in the system. In addition, following the suggestion, the sizes of numbers and characters within the figures have been enlarged in this revision.

  • Please review for the language, there are many typos and sentences with loss of meaning due to grammatical errors. In many cases the plural suffix is missing (for example in the title: …different conditions)

Response: We greatly appreciated for your suggestions. The whole manuscript has been double checked thoroughly. The typos and the grammatical errors have been carefully corrected (English editing by MDPI, Certificate-60654). All the corrections have been marked with blue in this revision.

  • Line 30. Use neurodegenerative diseases, not „neural degenerative diseases”

Response: Many thanks to your kindly suggestion. As you suggested, “neural degenerative diseases” has been replaced by “neurodegenerative diseases” in line 30 and marked with blue color.

  • Line 71. Are NPCs cultured in the form of neurospheres? Explain the method more clearly.

Response: Thank you for the question. Yes, NSCs/NPCs were cultured in the form of neurospheres according to the standard protocol. Following your suggestion, the description of method in detail has been added into the revised manuscript “5.1. Prepare of NSCs” and marked with blue color.

  • Line 104. Does the control group have medium with glucose?

Response: Thank you for the question. Yes, the medium used in control group that is the mechanical trauma model with glucose.

  • Line 106. How did you make sure that all slices had similar mechanical damage? Did you make specific number of holes with the pipette tip?

Response: Thank you for the questions. Regarding the mechanical trauma model, same numbers of holes in each slice were made by same person (the first author, Qian Jiao) in this study. The mechanical damage of each hole might be slightly different due to the manual operation.

  • Line 152. Process extension sounds strange, consider using another term such as neurite outgrowth

Response: Many thanks for the suggestion. The reason for using nonspecific word “process”, rather than “neurite” is that the phenotype of the NSCs daughter cells is not finally confirmed yet. They could be neuronal progenitors/neurons or other cell type. Therefore, the word ‘process’ has been remained. We made a relative revision by adding ‘the daughter cells’ following NSCs here and marked with blue color.

  • Co-localization of EGFP with some stainings in Figure 1.B is not clear. I do not know if it is from the low image quality in the draft, but if not would be good to put green and red channel also separately if not (such as MAP2, not clear if the transplanted cell is positive or is it from the slice)

Response: We are sorry for the difficulties in reading the images. Figure 1B has been revised as you suggested. Two separately channels with green and red were provided in the revised manuscript. The original figures also have been uploaded in the system. 

  • Fig.1C – please indicate which brain region this is. Is the statement with Ki-67+ cells correct for all the brain regions?

 Response: We are very sorry for the confused description. Brain slices of forebrain in this study were isolated from the newborn rats (postnatal 3 days). There are a small portion of cells/progenitors still keeping proliferation (Ki-67+). Our previous study also showed the nestin positive cells exist in cerebral cortex and striatum of newborn rats at postnatal 3 days.

Reference:

Jiao, Qian, Xie, Wuling, Wang, Yuanyuan, Chen, Xinlin, Yang, Pengbo,  Zhang, Pengbo, Tan, Jing, Lu, Haixia, Liu, Yong. Spatial relationship between NSCs/NPCs and microvessels in rat brain along prenatal and postnatal development. Int J Dev Neurosci, 2013, 31(4): 280-285.

  • When talking about NSC migration, (such as figure 3G) do the authors indicate the cells are migrating due to their shape, or is there an original location where the cells were transplanted and they were observed later outside of this location? When the cells are transplanted, they can also move on the surface of the slice due to the shape of the brain slice as the edges typically get thinner. Or the parts with mechanical damage can differ in topology, causing not equal distribution of the NPCs. Do the authors account for these? Zoomed out images such as Morgan et al., 2012; Fig3.A can be good to show there is no difference.

 Response: We appreciated very much for the professional questions. In the current study, ‘NSCs migration’ indicated that the grafted NSCs were observed later outside of the original location where they were transplanted. Regarding the thickness of distinct brain regions, the differences do exist. We noticed that within the slice, cerebral cortex was thicker than striatum after cultured in vitro for 7 days, and the mechanical damaged part was thinner than the surrounding parts, as you mentioned. In order to reduce the effects of topology differences, we transplanted NSCs on the whole slices surface, as showed in Figure 3A (the schematic diagram of transplantation of NSCs on OGD brain slices), and then conducted a long-term observation (more than 30 days) to observe the differences in the same region at different time points. We totally agree that the zoomed out images will be better to show the difference. It is with great regret that we had not taken the zoomed out images of Figure 3G. We will follow this suggestion in our following experiments.  

  • Fig.4C what does the dashed white line mean? Is it the area of injury? Please specify.

 Response: Thanks for the question. The dashed white line was the boundary between normal and traumatic area. The characters of “traumatic area” were marked in the merged images in Figure 4C.

  • Line 277. Co-cultured NSCs survive better etc… Would be good to specify NSCs co-cultured with organotypic slices, as they can also be co-cultured with other type of cells.

Response: Many thanks for the suggestion. ‘Co-cultured NSCs survive better’ in this manuscript means ‘NSCs co-cultured with organotypic slices’. The viablity of these cells are better than cultured alone. Our previous work and other peer researches also showed the same results that, compared with monocultures, co-cultured human neural precursor cells with brainstem slices survive and migration better [26].

References in manuscript:

  1. Novozhilova, E., et al., Neuronal differentiation and extensive migration of human neural precursor cells following co-culture with rat auditory brainstem slices. PLoS One, 2013. 8(3): p. e57301.

  • Authors mention guidance cues many times, could be good to discuss in one/a few sentences what can be these, in which way they can different in the brain regions evaluated in this study.

Response: We greatly appreciate the reviewer’s comments. Yes, it would be better to clarify what are the cues in different brain regions. They could be the cytoarchitecture, extracellular matrix or the soluble signals. Following the suggestion, few sentences have been added to the discussion and marked with blue color (line 229-232).